# DNA nanomapping using CRISPR-Cas9 as a programmable nanoparticle

Andrey Mikheikin[1], Anita Olsen[1], Kevin Leslie[1], Freddie Russell-Pavier[2,3], Andrew Yacoot[2], Loren Picco[3], Oliver Payton[3], Amir Toor[4,5], Alden Chesney[5,6], James K. Gimzewski [7], Bud Mishra[8] & Jason Reed[1,5]

Progress in whole-genome sequencing using short-read (e.g., <150 bp), next-generation sequencing technologies has reinvigorated interest in high-resolution physical mapping to fill technical gaps that are not well addressed by sequencing. Here, we report two technical advances in DNA nanotechnology and single-molecule genomics: (1) we describe a labeling technique (CRISPR-Cas9 nanoparticles) for high-speed AFM-based physical mapping of DNA and (2) the first successful demonstration of using DVD optics to image DNA molecules with high-speed AFM. As a proof of principle, we used this new "nanomapping" method to detect and map precisely BCL2–IGH translocations present in lymph node biopsies of follicular lymphoma patents. This HS-AFM "nanomapping" technique can be complementary to both sequencing and other physical mapping approaches.

[1] Department of Physics, Virginia Commonwealth University, Richmond, 23284 VA, USA. [2] National Physical Laboratory, Hampton Road, Teddington, TW11 0LW Middlesex, UK. [3] Interface Analysis Centre, H. H. Wills Physics Laboratory, Tyndall Avenue, Bristol BS8 1TL, UK. [4] Department of Internal Medicine, VCU School of Medicine, Richmond, 23284 VA, USA. [5] VCU Massey Cancer Center, Richmond, 23284 VA, USA. [6] Department of Pathology, VCU School of Medicine, Richmond, 23284 VA, USA. [7] Department of Chemistry and Biochemistry, UCLA, Los Angeles, 90095 CA, USA. [8] Departments of Computer Science and Mathematics, Courant Institute of Mathematical Sciences, New York University, New York, 10012 NY, USA. Correspondence and requests for materials should be addressed to J.R. (email: jcreed@vcu.edu)

Among the most important applications of DNA physical mapping are structural variant detection and whole-genome haplotype assembly. Because of several earlier probabilistic analyses (including our work on 0–1 laws[1–3]), it has become clear that successful physical mapping technologies will need to exceed certain thresholds in molecular length, coverage, sizing accuracy, and marker specificity. Given this design requirement, the field urgently needs quantum leaps in both super-resolution-microscopy technology and labeling chemistry.

Despite being invented in the mid-1990s, single-molecule optical mapping has had only a limited impact on the field of human genomics compared to the more recently invented next-generation sequencing, primarily because it lacks sufficient resolution to deal with structurally complex regions easily. We have shown previously that marker localization must be extremely precise (e.g., uncertainty <250 bp) to make single-molecule alignment computationally straightforward[1–3]. While very large contiguous maps can be generated with optical mapping, the error in localizing individual features is large enough (e.g., ~2–5 kb) that relatively high-coverage depth (e.g., >100×) is required to make it minimally workable. Even with this high coverage, optical mapping often fails as a stand-alone method for assembling high-complexity genomes. It is for this reason that the most successful applications of that technique in human genomics have been as a supplement to sequencing approaches (e.g., Illumina, PacBio, etc.)[4] It is possible that in time, "super-optical-resolution" techniques (e.g., PALM, STORM, etc.) will transform optical mapping. However, recent attempts to employ these methods rely on protocols that remain to be proven as practical for general use (lengthy, complex, etc.)[5–7].

Nanopore technology, in theory, is capable of both base-by-base sequencing, and long-range mapping of DNA structural variants. However, the physics of the DNA–nanopore interaction present some significant barriers to analyzing long molecules, and there are distinct advantages to working with longer, intact molecules for haplotype phasing and bridging repeats. Unlike optical mapping, nanopore techniques employ a "fixed sensor,

mobile molecule" approach. Because long mobile DNA molecules form random coils in solution, slow diffusion rates, and molecule tangling are barriers to achieving long reads. While very long nanopore reads (e.g., 50 kb+) have been demonstrated sporadically, this performance has proven to be difficult to achieve routinely. Further, these very long reads required library preparations and have not been achieved with unmodified genomic DNA. Recent work shows that nanopore DNA sequencing is most effective when using molecules of <10 kb in length[8, 9]. While this read length is considered to be long in the next-generation sequencing field, it is quite short for a physical mapping method. In comparison, the "imaging" approach to DNA mapping (e.g., optical mapping[10], fiber FISH[11], DNA combing[12], etc.) uses fixed, extended molecules. This lends itself well to analyzing natural genomic DNA fragments of great size. For instance, optical mapping easily handles single molecules as large as ~300 kb[13]. Our work with AFM imaging has taken the "fixed molecule" approach, adopting the reproducible and well-known imaging-field protocols that are compatible with very long, surface-supported molecules[2, 14].

In this work, we use high-speed AFM (HS-AFM) to create precise DNA single-molecule physical maps, and show scalability by integrating consumer electronics (DVD optical pickup units). To date, the impetus behind the field of high-speed AFM development has been the visualization and study of biomolecular processes. The instruments designed to achieve this goal all sacrifice sample versatility to provide a small (e.g., 1 × 1 μm) window into the nanoscale[15, 16]. The lack of scalability of these measurements prohibits their use in physical mapping, which requires both high-data rates and wide-area coverage. In contrast, our method for high-speed AFM directly addresses this need. It follows design principles distinct from the rest of the field: operating in a high-speed contact mode, we sidestep bandwidth bottlenecks and enable unprecedented rates of nanoscale measurements.

To unlock the potential of HS-AFM-based "nanomapping", we developed Cas9 as a stable and specific "programmable

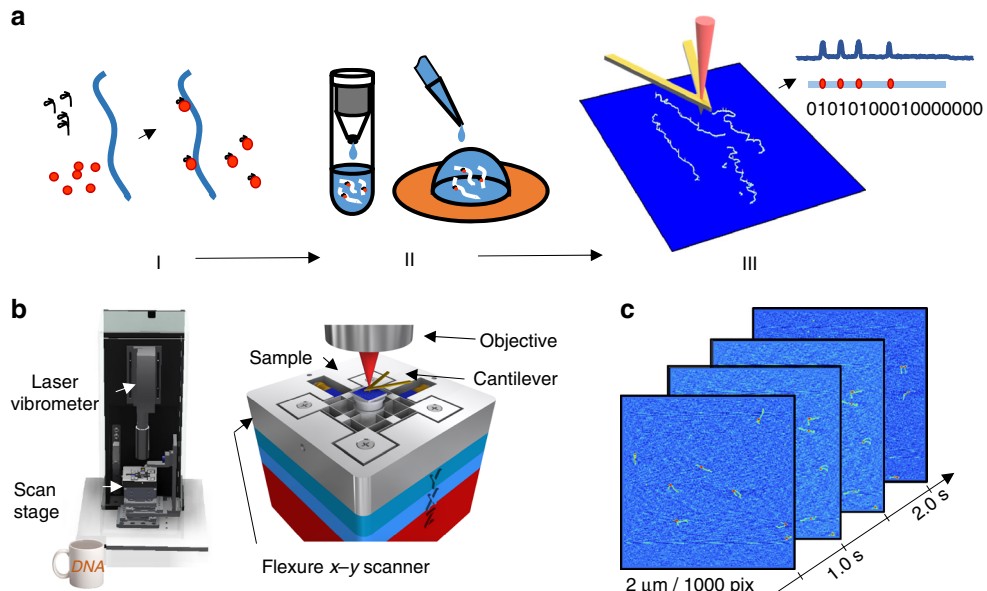

**Fig. 1** High-speed AFM measurement of Cas9-labeled DNA. **a** Cas9 labeling chemistry and sample preparation work flow: (I) incubate target DNA with Cas9–sgRNA and fix with formaldehyde, (II) remove unbound Cas9–sgRNA complexes and deposit molecules on mica, and (III) image with HS-AFM and measure locations of labels on DNA. **b** High-speed contact mode AFM, with a coffee mug for scale. The zoom-in view of the x–y flexure scanner shows the relative positions of the sample, cantilever, and laser vibrometer objective. **c** As configured, the HS-AFM produces two 16-bit, 1000 × 1000 pixel images per second

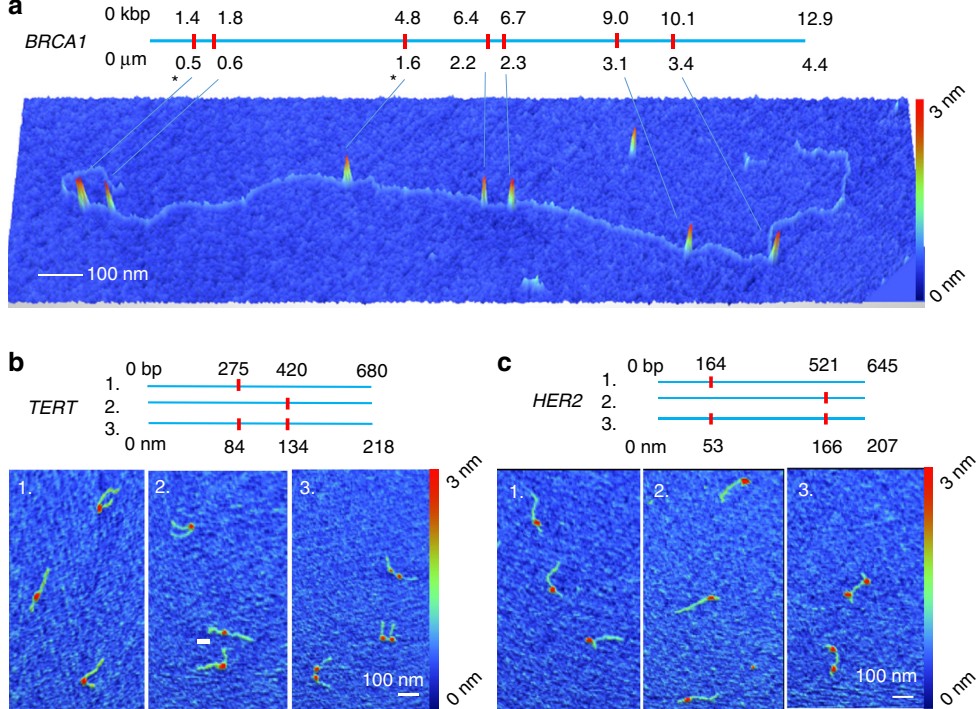

**Fig. 2** sgRNA–Cas9 complexes on DNA. **a** Map and the corresponding HS-AFM image of Cas9-labeled *BRCA1* amplicon. Red tick marks on the map indicate locations of the Alu repeat-targeted sgRNA-binding sites. In the image, sgRNA–Cas9 protein complexes appear as 3-nm-tall bumps on the DNA backbone. The amplicon sequence includes five perfect matches to the 20-bp sgRNA sequence and five single base-pair mismatches. Single base-pair mismatches label at roughly one-half the frequency of perfect matches. The star (*) indicates Cas9 bound at locations with a single mismatch, while all other locations are perfect matches. **b**, **c** Cas-9-labeled *TERT* and *HER2* amplicons. The sharp bend induced in the DNA backbone by the sgRNA–Cas9 complex is evident in both examples

nanoparticle". One very discovery, we report is the relative stability of the Cas9–sgRNA–DNA complex in the face of the harsh perturbation generated by the AFM tip moving at linear speeds of up to 10 mm/s. Previous work on DNA labeling with CRISPR-Cas9 has focused on its properties and uses as an enzyme, which is very different from our use of it as a "nanoparticle". For instance, in a recent optical mapping study, it served as a nicking restriction enzyme, to create sites for secondary DNA polymerase labeling with fluorescent nucleotides[17]. Molecule fragmentation is a major drawback of nicking-based labeling schemes. DNA with closely spaced nicks (e.g., less than ~200 bp) is not stable, and will not remain intact during processing. This problem impairs precisely locating translocation breakpoints, or localizing short insertions or deletions. It also puts an upper bound on the maximum length of the molecule, because very long molecules will inevitably contain two nearby nicking sites. Using Cas9 as a nanoparticle instead of a nicking enzyme avoids these drawbacks[15–17].

## Results

**Cas9 nanoparticle labeling.** Figure 1a shows the Cas9 labeling chemistry and sample work flow. Target DNA is incubated with Cas9–sgRNA and fixed with formaldehyde to prevent dissociation of Cas9 from DNA during purification. Unbound Cas9–sgRNA complexes are removed, and the sample is deposited on mica for HS-AFM imaging in air. Prior to imaging, the sample is dried at 120 °C, a step which improves imaging performance and also likely reverses the formaldehyde fixation[18]. During imaging, the cantilever position is measured with a Polytec compact laser Doppler vibrometer capable of capturing 2.5 million height measurements per second with a noise floor below 200 pm. Piezo stick-slip motors below the scanner can

provide coarse positioning in x, y, and z, and permit imaging anywhere within a 17 × 17-mm area of the sample. It can produce two 16-bit, 1000 × 1000 pixel images per second. Scan sizes are 0.5–5-microns square (2 microns typical), yielding 0.5–5-nm pixel sizes.

We used a positively charged surface for DNA fixation, as the behavior of this type of system in AFM imaging has been studied extensively[19–21]. Although several methods can be employed to charge the surface, and would work for our application, magnesium ion-mediated deposition on freshly cleaved mica is simple, rapid, and allows for accurate measurement of DNA contour length. As shown by refs. [20, 22], the binding forces are strong enough to irreversibly adsorb DNA molecules onto mica surface but at the same time allowing for equilibration of DNA molecules in two dimensions on the surface.

We demonstrated the efficiency and precision of Cas9 labeling using several gene-specific sgRNAs targeted to *BRCA1*, *HER2*, and *TERT* gene sequences (see Supplementary Fig. 1 for maps). Cas9 formed sequence-specific complexes with sgRNA at high efficiency (e.g., median 90% per site labeling rate), and labels were localized on individual molecules with very high precision. Figure 2a shows HS-AFM images of complexes of wild-type Cas9–sgRNA with *BRCA1* long-PCR amplicons (12,900 bp). Bound globular particles are present on the DNA molecules in close proximity to the expected target sequence locations. Note that no significant cleavage of DNA by wild-type Cas9 nuclease at our experimental conditions was observed. To label the *BRCA1* amplicon, we employed a single sgRNA that matches a subset of naturally occurring *Alu* repeats in the genomic locus (see Supplementary Fig. 2 for further exemplary images). In this case, the labeling rate was ~90% for perfect sgRNA match sites and 50% at sites with a single mismatch. *Alu* repeats are the most

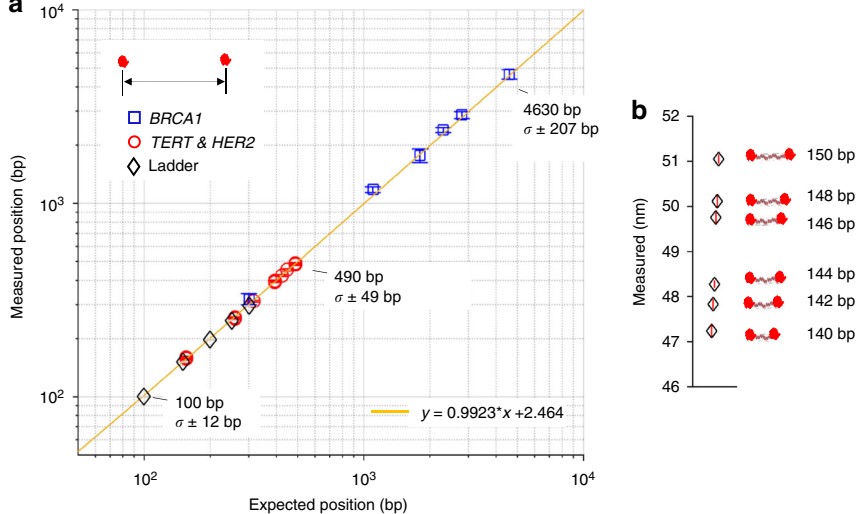

**Fig. 3** HS-AFM accurately localizes sgRNA–Cas9 complexes on DNA. **a** Measured vs. expected position of the Cas9 labels determined for *BRCA1*, *TERT-HER2*, and ladder constructs. Error bars represent the 95% confidence interval of the mean. For the *TERT-HER2* and ladder data points, the symbol diameter is larger than the 95% confidence interval span. The expected size and sample standard deviation ($\sigma$) are indicated at three points for reference. **b** Sizing closely spaced ladder constructs, end-labeled with Cas9. Vertical red lines inside symbols represent the 95% confidence interval of the mean. See Methods for measurement detail

abundant repetitive element in the human genome, making up 10% of the total nucleotide sequence. As such, *Alu* repeats are convenient, densely spaced targets for CRISPR-Cas9 mapping. For example, 42% of the tumor suppressor gene *BRCA1* sequence consists of *Alu* repeats[23]. Moreover, *Alu* repeats mediate some cancer-related genome variations, including deletions or duplications of exons in *BRCA1*[24–28].

Figure 2b, c show sequence maps and the corresponding HS-AFM images of singly- and doubly labeled *TERT* (680 bp), and *HER2* (645 bp) constructs. Custom sgRNAs were created to label these constructs for the purposes of accurately determining positioning accuracy and labeling kinetics. For the *TERT* amplicon, two unique sgRNAs were tested independently and in combination; for *HER2*, five unique sgRNAs were tested independently and in combination. The labeling was highly efficient, with over 90% of the expected sites containing a Cas9 label (median 94%, range 89–99%). One sgRNA species exhibited an unusually low initial labeling rate (35%); this was improved to 87% by increasing the sgRNA concentration to threefold vs. standard conditions. This result suggests that many DNA sequences containing a GG motif can be targeted with proper reaction optimization. Off-target binding, defined as labels >2 s.d. from their expected location, was quite low and did not exceed 6% on a per-site basis. However, the off-target binding rate depends on experimental conditions (i.e., concentration of Cas9 and sgRNA) and can also be optimized. Labeling efficiency values for the different sgRNAs are presented in Supplementary Table 1.

The spatial resolution of labeling, i.e., the precision of location of the label on DNA molecule, is a critical parameter in DNA mapping (see e.g., Anantharaman et al.[1] or Anantharaman et al.[3]). As shown in Fig. 3a, the HS-AFM positional measurements were highly accurate in an absolute sense and linear over the measured range of 100–4550 bp. In addition to long-range mapping applications (e.g., sequence assembly, detecting structural variants, etc.), there are many potential applications requiring precise sizing of short DNA molecules (e.g., smaller than 300 bp). For example, short-tandem repeat-based genetic fingerprinting requires the ability to distinguish 2–4 bp length differences in PCR amplicons of ~200 bp total length. To test the ability to distinguish such small size differences, we measured a

ladder series of 140–150 bp PCR amplicons, labeled at both ends with Cas9. As shown in Fig. 3b, we reliably discriminate 2 bp differences in these amplicons by pooling measurements from a large number of identical molecules (~15,000 counts per species). Notably, the image quality was sufficiently good to allow online, real-time image processing. We are not aware of another AFM technology with the throughput required to make these kinds of measurements in a timely fashion. Other potential applications include measuring spatial properties and conformational changes of A-, B-, Z,- and other forms of duplex DNA and noncanonical DNA structures such as triplexes, quadruplexes, and four-way junctions of various sequence composition at different experimental conditions (pH, temperature, ions, etc.). Cas9-labeled DNA molecules could be employed as a highly sensitive biosensor to detect the presence of DNA-specific agents (small molecules, DNA adducts, DNA-specific proteins, histones, etc.).

**BCL2–IGH translocation mapping.** An important potential application of "nanomapping" could be detecting cancer-related structural variants of diagnostic and prognostic significance. In the clinical lab, fluorescent in situ hybridization (FISH) and PCR remain the mainstays[29]; unfortunately, they fail in a significant fraction of cases, either due to insufficient resolution (FISH) or the fact that the vast majority of structural variant breakpoints are scattered widely and thus cannot be localized a priori for amplification by PCR[30]. While microarrays can improve the detection of copy number variations, they are not a replacement for FISH, for example, because they cannot detect unlocalized balanced translocations.

As shown in Fig. 4, we assessed the feasibility of using Cas9-HSAFM mapping to detect the *BCL2–IGH* translocation associated with follicular lymphoma, caused by aberrant V(D)J recombination[31, 32]. We chose this example because it is common in lymphoma, and likely to be present in easily obtained frozen tissue shavings. The breakpoints for this translocation are well characterized and serve as a reference for validating the accuracy and reproducibility of the results using this technique in clinical samples, such as bone marrow and lymph node biopsies. These breakpoints occur adjacent to the Ig heavy-chain J-region ($J_H$) or adjacent to the junction of the heavy-chain D-region and J-region

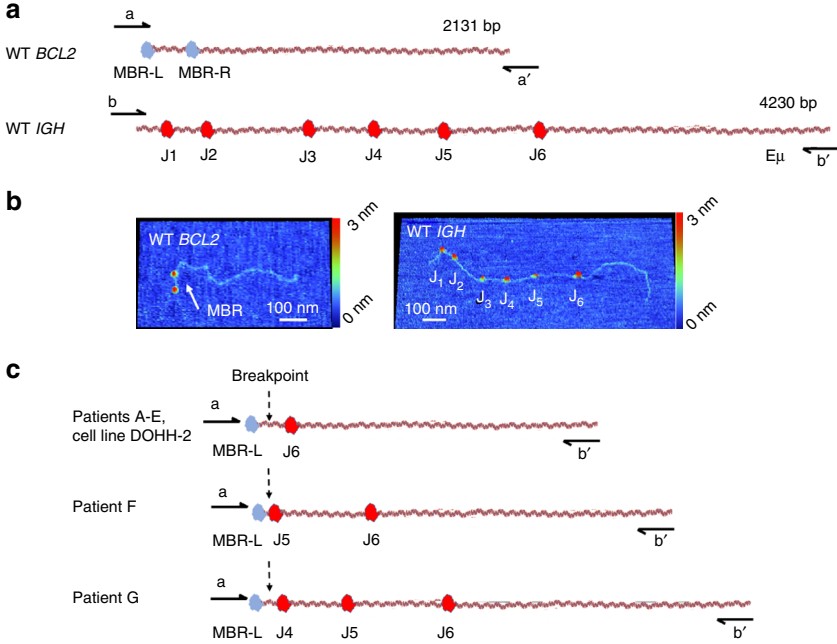

**Fig. 4** Mapping BCL2–IGH translocations with Cas9. **a** Maps of PCR amplicons from wild-type BCL2 and IGH with Cas9 labels (different colors for emphasis). The two labels on *BCL2* straddle the 300-bp MBR translocation hotspot. **b** HSAFM images of labeled amplicons. **c** HSAFM-determined maps of the translocation breakpoints from a follicular lymphoma cell line DOHH2, and the seven patient samples. Identity of the Cas9 labels is determined by measuring their position from the end of the molecule. Maps were confirmed with 10 or more molecules each. In all seven patent samples, the translocation breakpoint mapped to the major breakpoint region (MBR) of BCL2 and the $J_H$ region of IGH

($D_H J_H$). The majority of the corresponding breakpoints in *BCL2* (e.g., about 60% of cases) occur in a hotspot known as the major breakpoint region (MBR)[30], making this region a convenient target for our tests.

We designed sgRNAs targeting the $J_H$ elements in *IGH* and the MBR region of *BCL2*. Figure 4a shows the labeling scheme and the corresponding images of wild-type *BCL2* and unrearranged *IGH* obtained from human germ line DNA. Using the Cas9 labels, we mapped a known *BCL2–IGH* translocation present in the follicular lymphoma cell line DOHH-2, obtained commercially, and detected similar translocations in seven lymph node tissue samples from follicular lymphoma patients (Fig. 4c). In doing so, we employed very closely spaced markers (tens of nanometers) that are well distinguished—a capability beyond what is possible with any form of optical mapping, including super-resolution optical methods[33, 34].

**Imaging DNA with DVD optical pickup unit-enabled HS-AFM**. We demonstrated the scalability of HS-AFM design by replacing the laser vibrometer, by far the most complex hardware component, with an off-the-shelf DVD player optical pickup unit (OPU)[33]. The OPU detector and its placement above the scan stage is shown in Fig. 5a, b, respectively. The OPU module uses an astigmatic focusing scheme to sense the vertical displacement of the HS-AFM cantilever: the image of the tightly focused laser spot changes the shape on the quadrant photodiode detector as the cantilever is moved vertically (see Supplementary Fig. 4). Nanometer displacement sensitivity is achievable by the OPU without modification, using a simple battery power source to minimize electronic noise. The OPU has onboard voice coil actuators to focus the detection beam, and possesses integrated laser control electronics and high-bandwidth (e.g., >10 MHz) signal amplifiers, making it ideal for rapid, high-precision displacement sensing. The OPU HS-AFM produced images of similar size, resolution, and quality as the vibrometer, although at

a lower frame rate (0.5–1 frames per second (fps)). As shown in Fig. 5c, d, the Cas9 labels are clearly resolved, as is the DNA backbone in some cases, indicating that the noise floor is at or slightly below the apparent level of the DNA height under these conditions (roughly 300 pm). We determined Cas9 intra-label spacing on doubly labeled TERT and HER2 amplicons with accuracy and precision was equivalent to that achieved with the laser vibrometer (Fig. 5e). The median center-to-center spacing between the two TERT markers was 47.7 nm (s.d. 8.1 nm) vs. 46.9 nm expected; and for HER2, it was 108.2 nm (s.d 17.1 nm) vs. 114.1 nm expected.

With no modifications, this OPU HS-AFM can be used to detect and measure length polymorphisms between closely spaced markers (e.g., closer than about 400 bp). Alternatively, multiple Cas9 labels can be used as simple single-molecule barcodes, making the technology useful in counting applications such as detecting gene copy number variation, digital PCR, or transcriptional profiling. Achievable improvements in signal-to-noise and drift compensation[34] will allow the OPU to reliably detect the DNA backbone as well as the Cas9 labels. Moreover, OPUs have typical detection bandwidths of 80–400 MHz (e.g., BluRay), meaning that if higher sampling rate digitization electronics are used, the pixel rate of an OPU-based instrument can be increased from 2 MS per second used here to as high as 80 or even 400-million samples per second, depending on the OPU model used. By increasing the scan speed and image size, these higher pixel rates can be translated into measurement throughput directly.

## Discussion

The nanomapping technique described here can be complementary to both sequencing and other physical mapping approaches. It will be necessary to develop workable protocols to label and elongate molecules larger than 13,000 bp, if nanomapping is to become practically useful for de novo structural variant detection and sequence alignment applications. In principle, one

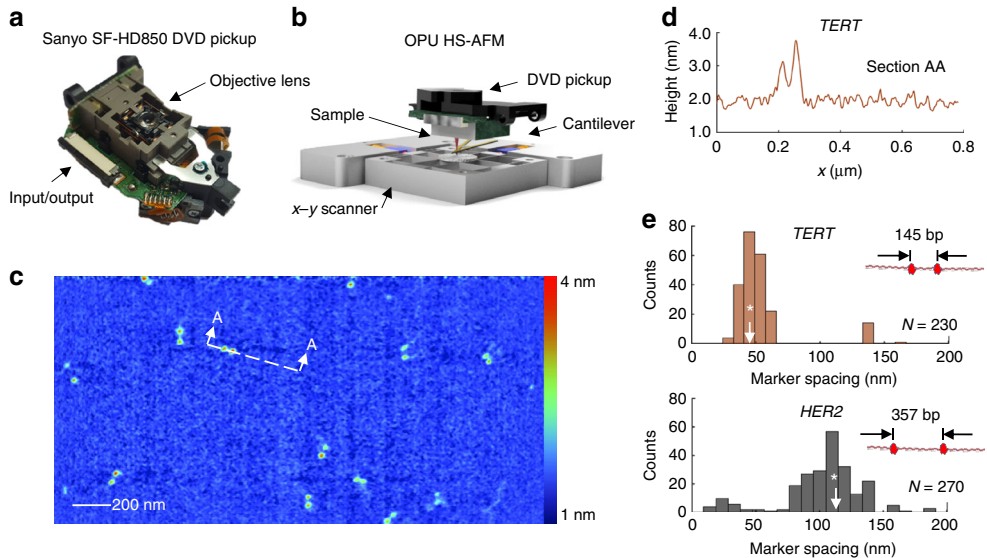

**Fig. 5** Detecting Cas9 labels with a DVD player part used as a displacement sensor. **a** Photo of the DVD player optical pickup unit showing the location of the laser-focusing objective and the signal and control connection point. Behind the objective lens is the 650-nm-wavelength diode laser, quadrant photodiode, and associated optics. **b** Schematic of the HS-AFM scan stage with the DVD optical pickup used as the cantilever displacement sensor. **c** An example image of *TERT* amplicons labeled with two Cas9 proteins. This image was obtained by averaging five full frames collected at the maximum frame rate of 2 frames per second. **d** Typical height cross section from double-labeled molecules obtained with the DVD optical pickup. **e** Histograms of Cas9 marker spacing measured for populations of *TERT* and *HER2* molecules. The white asterisk indicates the expected value

could isolate longer molecules by methods such as hybrid capture, gel extraction, centrifugation, immunoprecipitation, etc. These longer DNAs could be stretched linearly for genomic mapping using well-established DNA elongation methods such as micro- or nanochannels, pressure and electrokinetic flow devices, or laser and magnetic tweezers.

It may be possible to combine HS-AFM nanomapping with hybrid capture or "inverse" long PCR, for instance, to isolate translocations breakpoint regions using only limited knowledge of the loci involved in the translocation[35, 36]. Breakpoint position and identity of the translocation partners would be resolved by "molecular barcoding" with CRISPR-Cas9. In the *BCL2–IGH* mapping experiments, we did not examine the histology of the lymph node tissues, so we do not know the fraction of B cells harboring the translocation. This sample type would be expected to contain a significant number of cancerous cells, all of which would have the translocation. We used long-range PCR to prepare the DNA molecules and thus, only the translocated locus would be amplified. Under these conditions, we would expect to detect even relatively rare instances (e.g., cancer cells present at 1% abundance or greater).

For loci that can be effectively sampled with long PCR, or with an ensemble of long PCR amplicons, this strategy is likely to be effective in detecting rare species present at 1% abundance or greater. Hybrid capture methods combined with throughput-optimized hardware and software are likely to be the most generally effective approach. This is in fact the method used for most NGS diagnostic panels and whole-exome sequencing assays (e.g., Agilent SureSelect capture baits, etc.). In that case, both mutant and normal sequences would be present in the sample, and thus, the limit of detection becomes a function of sampling depth. Given the throughput achieved in this study, it is reasonable to project that the loci of several hundreds of kilobases can be sampled at a sufficient depth to detect mutants present at 10% fraction.

The data presented in Fig. 3 or Supplementary Tables 4–7, allow us to estimate the trade-off between time and resolution with the system, as implemented. For a single 12.9-kb BRCA1

molecule, the inter-Cas9 spacing is measured to a median precision of roughly ±100 bp s.d. The field of view is roughly 2.5 μm × 2.5 μm, and the achievable frame rate is 4 fps. Two overlapping frames are required to capture the entire BRCA1 molecule. This gives a very rough estimate of the time required to capture one such molecule at ±100-bp marker resolution: 0.5 s. For smaller molecules (100–300 bp), the median resolution for a single molecule is 20 bp. It is typically possible to capture between 10 and 100 such molecules per frame.

Throughput could be increased by (1) increasing the number of molecules per frame, using various flow alignment techniques to increase packing density, (2) expanding the field of view to the maximum supported 5 × 5 μm, and increasing the digitization rate accordingly, to retain pixel size, and (3) employing multiple cantilevers simultaneously on the same sample. A multiprobe cantilever chip is not strictly required. Since only a tiny fraction of the area of the sample is interrogated, it can be easily split and measured on spatially separated detector–cantilever pairs, assuming that the detector and digitizer used is relatively inexpensive, such as is the case for the DVD OPU.

While DVD OPUs have been used as displacement sensors in AFM previously, the combination with high-speed AFM presented here is new. An examination of the state of the field suggests that in a number of respects, our implementation exceeds the current state of the art. Our passive mechanical feedback implementation of high-speed "contact mode" with OPU detection is unique, as are the much higher imaging rates achieved, and the much larger measurement areas demonstrated. Because of these features, our DVD OPU implementation uniquely enables a large number of measurements of DNA molecules, which in turn makes possible entirely new statistical analyses of molecule conformation, counting rare species, precision length measurements, etc.

Although DVD optics were first implemented in AFMs in 1999[33] the field of high-speed AFM is much more recent and here, we present the results collected at a speed 10,000× faster than comparable implementations[37]. In a previous publication, it was shown that OPUs can act as a detection system in AC or

"tapping" mode AFM to image DNA with scan rates of 0.2 lines per second at pixel dimensions of $256 \times 256$[37]. In comparison, the method that we presented could perform AFM on a unit area to matching pixel densities 10,000 times at the same time compared to the previously reported technique.

In a separate publication by the same group[38] detailing improved scan rates, tip velocities are achieved of up to 1.4 mm/s. This compares well to our presented method which achieves scan rates of up to 16 mm/s. However, this previous work utilized an unusual 5-MHz-frequency cantilever, whereas our system operates with a mass-produced contact-mode cantilever. In addition, in the prior work, the stage is specified to have a maximum measurement area of up to 17 μm by 17 μm, whereas with our presented method, a 2 cm by 2 cm area is addressable. This improvement corresponds to a 700,000 times increase in measurement area.

Finally, the previous OPU implementations utilizing tapping or AC mode AFM[39] are inherently rate limited by the oscillation of the cantilever to pixel rates at a fraction of the oscillation frequency (typically <500k pixels per second). Therefore, in these implementations, it is not possible to access OPUs' ultrahigh bandwidth, capable of measurement rates up to several 100 s of MHz, for high-speed AFM. While with our presented method of high-speed contact-mode AFM, with no reliance on the resonant frequency, no such mechanical upper limit has yet been found; in our case, the limit is set by the digitization of the detection system. Therefore, this technology presents an exciting possible opportunity to increase high-speed AFM to even higher physical measurement rates, allowing for both better spatial resolution and frame rates.

While this tremendous speed increase is an essential difference, it is not the only one. We operate our instrument in a water-layer-mediated imaging mode that is unique to our instrument and critical to making measurements at very high scan rates[40]. Finally, here, we report how by combining large-travel coarse positioners and small high-speed scan stage, the technology enables measurements to be rapidly collected from the entire surface of the sample, rather than the much smaller areas typical of AFM. These design decisions make our instrument stand out, to the best of our knowledge, against the body of AFM development to date.

## Methods

**Reagents**. All primers and sgRNA were purchased from Integrated DNA Technologies. For sequences, see Supplementary Table 2. SgRNA was prepared according to the manufacturer's recommendations. Wild-type Cas9 protein was purchased from New England Biolabs, and Cas9 (D10A) nickase and dCas9 (D10A and H840A) were purchased from Novateinbio.

**DNA constructs**. Supplementary Fig. 1 shows maps of the constructs indicating the position of the Cas9 labels. For *HER2* and *TERT*, primers were designed with Primer Blast software, while primer sequences for *BRCA1* were taken from Jia et al.[41]. PCR solutions for *HER2* and *TERT* amplicons contained 200 nM primers (each), 100 ng of human genomic DNA: female (Promega) and 1× of LongAmp Hot Start *Taq* 2× Master Mix (New England Biolabs) in 20- micrl reaction volume. Amplification protocol for *HER2* and *TERT* was initial denaturation at 95 °C for 30 s, 35 cycles: 94 °C for 30 s, 60 °C for 30 s, and 68 °C for 2 min, followed by final extension at 68 °C for 5 min. For *BRCA1*, PCR solution composition and amplification protocol were the same as described in Jia et al.[41] (PrimeSTAR GLX DNA Polymerase, Takara). All amplicons were purified with Agencourt Ampure XP magnetic beads (Beckman Coulter Life Sciences), eluted in TE buffer and quantified either with Qubit dsDNA HS Assay or Qubit dsDNA BR Assay kit (Life Technologies).

For the *BRCA1* amplicon, *Alu*-family repeats containing the necessary GG sequence to be used as a PAM site were identified with RepeatMasker software. Using BLAST software, we found five perfect and five 1- or 2-base mismatched sites to the chosen sgRNA sequence (designated as Alu-sgRNA in Supplementary Tables 2 and 3).

As described in Supplementary Fig. 3, the ladder amplicons were synthesized with *Alu*-targeted sgRNA at the ends using probes to introduce the recognition and

primer sites. PCR solutions for ladder amplicons contained 400 nM of Ladder-Primer (the same primer is used as forward and reverse), 40 nM of Ladder-Probe, 1 ng of *BRCA1* amplicon, and 1× of LongAmp Hot Start *Taq* 2× Master Mix (New England Biolabs, Woburn, MA) in 20-micrl reaction volume. Amplification protocol for ladder amplicons was initial denaturation at 95 °C for 30 s, probe annealing, and extension at 50 °C for 5 min, 35 cycles: 94 °C for 30 s, 60 °C for 30 s, and 68 °C for 1 min followed by final extension at 68 °C for 5 min. The ladder amplicons were purified and quantified using the same protocol as other amplicons. Labeling was done with Alu-sgRNA using a standard labeling protocol.

**Labeling reaction**. The protocol is based on in vitro digestion of DNA with Cas9 protocol (New England Biolabs). EDTA was added to avoid cleavage of DNA, and the concentrations of Cas9 and sgRNAs were increased to improve the labeling rate. After labeling, Cas9-DNA complexes were fixed with formaldehyde, and the unbound protein was removed with a magnetic bead-based purification system. A concentration at 400 nM of each sgRNA was incubated with 200 nM of Cas9 protein in 1× Cas9 Nuclease Reaction buffer (New England Biolabs) supplemented with 50 mM EDTA (pH = 8.0) at room temperature for 10 min; next, amplicons were added to a final concentration of 5 ng/μl and incubated at 37 °C for 1 h; next, formaldehyde was added to a final concentration of 1% and incubated at room temperature for 10 min; next, TRIS (pH = 7.6) was added to a final concentration of 750 mM and the labeled amplicons were purified with Agencourt Ampure XP magnetic beads (Beckman Coulter Life Sciences). We used Ampure magnetic beads instead of spin columns, though both are effective. We had previously found contamination in some manufacturer's spin columns but not in Ampure, and thus gravitated to the latter[42]. The role of formaldehyde cross-linking is to prevent dissociation of Cas9 from DNA during purification, which precedes heating and imaging. We believe that it is very unlikely that Cas9 is covalently attached to DNA during HS-AFM imaging. Our deposition protocol includes 10 min incubation of deposited Cas9-DNA complexes at 120 °C followed by slow cooling, whereas formaldehyde cross-linking is reversed completely at 90 °C for 1 h or at 95 °C for 30 min[18, 43]. The role of formaldehyde cross-linking is thus to prevent dissociation of Cas9 from DNA during purification, which precedes heating and imaging. These data suggest the unique properties of Cas9 to form a stable, slowly dissociating tertiary complex with sgRNA and DNA.

**High-speed AFM sample preparation and imaging**. For HS-AFM sample preparation (all samples except *BRCA1*), imaging, and data analysis, we used a previously developed experimental protocol[2, 42]. One microliter of 100 ng/μl solution of the amplicons in deposition buffer (10 mM TRIS, 10 mM MgCl$_2$, pH = 7.6) was deposited on freshly cleaved mica surface, incubated for 1.5 min in a humid environment, rinsed three times with 200 μl of MQ water, baked at 120 °C for 20 min, and cooled down to 40 °C in an oven.

**Elongation of long DNA**. This procedure was used for the BRCA1 long PCR amplicons. A rectangular-shaped channel of 2–3-mm width was drawn on freshly cleaved mica with a marker. One-half of a microliter of labeled DNA in deposition buffer was deposited at the bottom of the channel, and the mica disk was immediately rotated by 180° along a horizontal axis, causing the sample droplet to flow over the surface due to gravity. The disk was washed three times with 200 μl of MQ water, baked at 120 °C for 10 min, and left in an oven until the temperature reached 40 °C.

**BCL2–IGH fusion sample preparation**. Follicular lymphoma cell line DOHH-2 was purchased from Leibniz-Institut DSMZ—Deutsche Sammlung von Mikroorganismen und Zellkulturen GmbH. De-identified follicular lymphoma patient samples were obtained from the VCU Tissue and Data Analysis Core Laboratory. Patient samples were obtained from patients who gave informed consent. This tissue was provided by the VCU Tissue and Data Analysis Core Laboratory (TDAAC), which collects residual tumor tissues and blood under an IRB-approved protocol (VCU IRB #HM2471). The sequence for MBR-02 and Eμ-01 primers was taken from Akasaka et al.[44], and other primers were designed with Primer Blast software. MagAttract HMW DNA kit (QIAGEN) was used to extract genomic DNA from clinical samples according to the manufacturer's protocol. The extracted DNA was quantified with Qubit BR or HS assays (Life Technologies). PCR solution contained 40 ng of template (clinical sample for gene fusion or human genomic DNA: female (Promega) for wild-type *BCL2* and *IGH* controls), 400 nM of each primer (for primer sequences, see Supplementary Tables 8 and 9), 1.6 μl of dNTP mix (2.5 mM of each in stock solution), 0.4 μl of PrimeSTAR GXL DNA polymerase (Takara Clontech), and 4 μl of 5x PrimeSTAR GXL buffer in 20-μl volume. Forty cycles: 10 s at 98 °C and 10 min at 68 °C were conducted (the protocol is based on Jia et al.[41]). Amplicons were purified with AMPure XP purification solution (Beckman Coulter) and quantified with Qubit BR or HS assays (Life Technologies). Labeling of gene fusions and control was performed with a standard labeling protocol with Cas9 concentration increased to 400 nM and MBR-BCL2-L gRNA was at 1200 nM.

**HS-AFM**. The system used in this work was a custom prototype which utilized a very low spring constant triangle cantilever (Bruker Nano, MSNL, 0.01–0.03 N/m

spring constant) operating in contact mode with no z-feedback. The cantilever vertical deflection was measured by a 2.5-MHz-bandwidth laser Doppler vibrometer (Polytec) using a height decoder module[45]. The sample was translated in the fast (1000 Hz) and slow (1–4 Hz) scan directions by a piezo-actuated flexure stage capable of 5 μm deflection in both axes. Typically, images of size $2 \times 2$ μm, $1000 \times 1000$ pixels, were captured in raster mode and rendered using customized LabView software[45]. The high-speed flexure stage was mounted on a stick-slip $x–y$ positioner for coarse alignment (Smaract, Inc.).

**AFM data processing**. An image-processing program developed for this application, called AFMExplorer, analyzed images[46]. The image-processing pipeline is as follows: AFM images are flattened and prefiltered to reduce noise, followed by adaptive thresholding based on pixel height to recognize regions corresponding to DNA molecules; a binary skeletonization procedure is used to determine the best backbone contour for each molecule, and the molecule length in nanometers is calculated by a cubic spline fit to the backbone pixel set. An additional feature was recently added for particle label recognition. This feature detects the height change on the backbone of the molecule due to each CRISPR label and calculates the length in nanometers of the distance from one end of the molecule to the label site. In the case of more than one label, the measurements are continued to each additional site.

BRCA1 amplicons spanned multiple overlapping frames; composite images were created before analysis by stitching individual frames into a single larger image, using the known displacement between frames. For HER2 and TERT molecules, in a minority of instances (e.g., fewer than 20% of images), the program was manually queued to correct or ignore automation errors; otherwise, the molecule identification and measurement is fully automatic. All ladder series measurements were fully automated.

Supplementary Tables 4–7 show measurement detail and statistics for the various constructs. The canonical solution value 2.94 bp/nm was used to convert contour lengths measured in nanometers. Values for the median and s.d. were determined from a Gaussian fit to the population data. For the BRCA1 data, lengths are measured between the center of the Cas9 labels appearing at perfect sgRNA match sites. For the TERT and HER2 data, lengths to the center of the Cas9 markers are measured from the molecule's unlabeled 5′ end (left end in Supplementary Fig. 1). The ladder constructs were labeled at the ends, and contour lengths were measured center to center between markers.

**Statistical analysis of labeling rate**. The full length of the molecule and the distance from one end of the molecule to the label of focus were used to bisect the molecule into a shorter and longer segment. The proportion of the full length represented by the segment of interest is the target sequence. The theoretical expected value of proportion can also be determined with this method using expected measurements of full length and site location. Proportions are bounded at 0 and 1, and are typically analyzed by transforming them into logit form defined as

$$\text{logit} = \ln\left(\frac{p}{1-p}\right)$$

The logit is the natural log of the long-segment proportion divided by the proportion's complement (one minus the proportion). This allows traditional linearized distribution statistics including means, standard deviations, and confidence intervals based on the logits. The means and the lower and upper limits of the logit-scale confidence intervals can then be back-transformed through exponentiation using

$$\hat{p} = \frac{e^{\text{logit}}}{1 + e^{\text{logit}}}$$

The percent of outliers for the 95% confidence interval of each label site is shown in Supplementary Table 1.

**Data availability**. All relevant data are available from the authors upon request.

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

## Acknowledgements

Funding for this work was provided by National Institutes of Health grant R01GM094388 to J.R., B.M. and J.K.G. and grant R01CA185189 to J.R. Services in support of the research project was provided by the VCU Massey Cancer Center Tissue and Data Acquisition and Analysis Core, supported, in part, with funding from NIH-NCI Cancer Center Support Grant P30 CA016059.

## Author contributions

J.R., B.M. and J.K.G. conceived the concept and planned the study, with input from A.Y., L.P., O.P., A.T. and A.C. J.R., A.M., A.O., F.R.-P., L.P. and O.P. carried out the experiments and analyzed the data. J.R., B.M., L.P., O.P., F.R.-P. and K.L. cowrote the paper.

## Additional information

**Competing interests:** The authors declare no competing financial interests.

