## [Peer Review File · Nature Communications]

Reviewers Comments:

Reviewer #1 (Remarks to the Author):

This manuscript reports effective gene mapping using atomic force microscopy (AFM). This is an obvious application of AFM and something my group made initial attempts at over ten years ago. The main difficulties are due to entanglement of long genomic DNA and a suitable morphological sequence marker. Moreover, throughput of AFM has been a major stumbling block for development of analytical techniques that can compete with other methods. This study does enough to demonstrate the significant potential advantages of faster scanning AFM for gene mapping at sufficiently high resolution.

1.1 We agree with the reviewer.

The success of this current work lies in the combination of using a wide scanner video-AFM system, a reliable gene mapping marker (CRISPR/Cas9) and the sample preparation for DNA constructs, where the molecules are baked on the mica support. This dehydration of the sample provides irreversible Van der Waals contact between the DNA and mica such that they are resistant to the shear forces exerted during contact mode AFM imaging. The authors promote the immobile molecule approach, which is no surprise to me, since my group has pioneered this for DNA-protein interactions for over a decade, while the bio-AFM field is often too pre-occupied with measuring everything in solution. The sample preparation technique appears to have developed sufficiently that the protein gene markers and DNA structure are not affected by dehydration and the DNA remains in B-form stabilised by the mica-DNA interactions. The inclusion of EDTA in the reaction mix to inhibit the nuclease activity of Cas9 is a simple and elegant way of turning it into a fiducial marker.

1.2 We agree with the reviewer.

Summary: The manuscript presents demonstration of reliable mapping on a 12.9kbp system as well as reliable distinction of two sequences separated by ~150bp. Furthermore, they demonstrate the use of the optical DVD pick-up set-up, presumably as a pre-cursor to developing simple lightweight technology to take this approach towards clinic, although the reasoning behind this is not particularly emphasised. It is a well written manuscript that shows all relevant data to demonstrate the effectiveness of the approach, but in some places it is written with a certain amount of hyperbole. Some modifications are required to clarify certain aspects before it is suitable for publication.

Specific Points to Consider;

1. In the introduction, the relative merits of different approaches to sequencing and physical gene mapping are discussed. In particular, they say the nanopore approach is limited typically to 10kb while optical mapping can go to 300kb but still has limited resolution. Here, they demonstrate mapping up to 12.9kb with resolution that currently beats optical but does not really surpass nanopores on length. What would be really good to know is the total time-scale they require to produce their maps with all positions accurately determined. Smaller separation of sequences requires higher statistics, but since these fragments are smaller perhaps the total acquisition time is similar. Is there a trade-off between resolution (through sampling) and larger DNA fragments ?

1.3 The data presented in Figure 3 and/or Table S4-1, allow us to estimate the tradeoff between time and resolution with the system, as implemented. For a single 12.9kb BRCA1 molecule the inter-Cas9 spacing is measured to a median precision of roughly +/- 100 bp. The field of view is roughly 2.5 um x 2.5 um, and the achievable frame rate is 4 fps. Two overlapping frames are required to capture the entire BRCA1 molecule. This gives a very rough estimate of the time required to capture one such molecule at +/-100 bp marker resolution: 0.5s. For smaller molecules (100-300 bp) the median resolution for a single molecule is 20 bp (see Fig 3 and/or Table S4-2). It is typically possible to capture between 10 and 100 such molecules per frame.

Throughput could be increased by: (1) increasing the number of molecules per frame, using various flow alignment techniques to increase packing density, (2) expanding the field of view to the maximum supported 5 um x 5 um, and increasing the digitization rate accordingly, to retain pixel size, and (3) employing multiple cantilevers simultaneously on the same sample. A multi-probe cantilever chip is not strictly required. Since only a tiny fraction of the area of the sample is interrogated, it can easily be split and measured on spatially separated detector-cantilever pairs, assuming the detector/digitizer used is relatively inexpensive. This is the motivation for using the DVD-OPU detectors (\$6 per cantilever detector, 80+ MHz digitization rate)

We have included this discussion in the revised manuscript.

2. The AFM system is run with no tip-sample feedback whatsoever. This would suggest that baking of the samples makes the DNA resistant to potentially large loading forces. It would be good to know if data collected during a mapping experiment is from only one sample without the need to re-engage the AFM tip at any stage, to control the tracking force.

1.4 There is no need to re-engage the tip, as the sample is level enough over the scan area. However an implementation with weak feedback (which does not limit scan speed), for more tilted samples, has been implemented as well.

3. Paragraph lines 59 to 67: describes relative merits of high speed AFM techniques but does not go into much detail, which would make it difficult for a general reader not familiar with all the ins and outs of AFM development to follow. The approach used here is based on the video-AFM technology developed by the Miles groups at Bristol, but this work is not cited. The authors should improve the clarity of distinction between the video-AFM approach and the high-speed AFM based on Ando designs, more clearly illustrating the advantage of their method.

1.5 We have added an extended discussion of the AFM system to the text. Please also see our answer #2.3, to a question by Reviewer 2.

4. The process for removing unbound Cas9-sgRNA complexes is not well explained except to say it's a magnetic bead system. More detail would be welcome. Is Fig 1a (II) a spin column ??

1.6 We used Ampure magnetic beads instead of spin columns, though both are effective. The graphic is only intended to capture the conceptual requirement for a purification step. We had previously found contamination in some manufacturer's spin columns but not in Ampure, and thus gravitated to the latter (Mikheykin et al, 2016). We have added this detail to the Methods.

A. Mikheykin, A. Olsen, L. Picco, O. Payton, B. Mishra, J.K. Gimzewski, and J. Reed, "High-speed AFM Reveals Contamination in DNA Purification Systems", *Analytical Chemistry*, DOI 10.1021/acs.analchem.5b04023, 2016

5) Is it not clear to what extent formaldehyde cross-linking is required for the AFM sample preparation? Does it work without this ? Does it cross-link the protein particle onto the DNA to make it resistant to degradation during baking ? Again, more detail would be useful.

1.7 The role of formaldehyde crosslinking is to prevent dissociation of Cas9 from DNA during purification, which precedes heating and imaging. We believe that it is very unlikely that Cas9 is covalently attached to DNA during HS-AFM imaging. Our deposition protocol includes 10 min incubation of deposited Cas9-DNA complexes at 120 C followed by slow cooling, whereas formaldehyde crosslinking is reversed completely at 90C for 1 h or at 95C for 30 min (see [Qiagen] and [Kaufmann]). This data suggests the unique properties of Cas9 to form a stable, slowly dissociating tertiary complex with sgRNA and DNA.

We have revised then methods description accordingly.

[Kaufmann] - Kaufmann *et al.* - "In organello formaldehyde crosslinking of proteins to mtDNA: Identification of bifunctional proteins", PNAS, 97, pp 7772-7777, 2000

[Qiagen] - "QIAamp-DNA-FFPE-Tissue-Handbook", June 2012

6) Some discussion of the DNA solution conditions for AFM sample preparation could be included with reference to work on DNA conformation on mica.

1.8 We have added this information to the manuscript.

Reviewer #2 (Remarks to the Author):

The authors prepared CRISPR/Cas9 for labeling specifically and efficiently targeted DNA sites, used AFM combined with DVD optics to localize the sites with a high precision and time-effectiveness. Examples include singly- and doubly-labeled TERT and HER2 constructs, and BCL2/IGH translocations. As far as I know, designing CRISPR/Cas9 of high specificity and efficiency (the labeling rate was 90% for perfect sgRNA match sites and 50% at sites with a single mismatch. And off-target binding was quite low.) is not an easy task at all. I appreciate highly such achievement. Also, I am very impressed by the accuracy of their approach (the ability to distinguish two-to-four base pair length difference in PCR amplicons of approximately 200 bp total length, Fig. 3b).

2.1 We agree with these comments.

However, I don't agree with many of key statements in the manuscript: 1) line 21, a key advantage of nanopore sensors is high parallelism. I don't think AFM can compete against the sensors in this aspect. The authors are comparing 'Apple' with 'Orange'.

2.2 We agree with the Reviewer that parallelism is a relative strength of commercial nanopore sensors; it was not our intention to compare HS-AFM vs nanopores on that basis. Rather, we bring up nanopores because the genomics community is interested in all emerging approaches which can address problems in detecting structural variants. We feel that our technology combines features which make it an attractive approach for high resolution physical mapping of DNA.

We have revised the original line 21 to clarify our intent.

We note that there are distinct advantages to working with longer, intact molecules for haplotype phasing and bridging repeats, where 'nanomapping' and optical mapping will likely outperform nanopore methods. Another is efficiency of sequence/map assembly, where due to the need for molecule overlap, shorter molecules must be sampled in greater depth, increasing cost and latency. One other relevant comparison is DNA throughput per cost, as the two technologies operate on similar time scales (minutes to hours). A single cantilever can interrogate multiple DNA molecules per frame (up to several hundred, depending on the molecule type). At four frames per second, the throughput of our system, measured as molecules or base pairs analyzed per second, is considerable. With the DVD+high-speed implementation, many parallel cantilevers can be added at low cost (about \$6 per DVD unit).

We have emphasized the above points in the revised text.

2) line 22, combination of DVD optics and AFM is already known. It is hard to understand for me why they claimed the approach as one of two substantial technical advances of their study. Therefore, the last paragraph and Figure 5 in the manuscript introducing the hardware design and its capability is absolutely redundant.

2.3 As the Reviewer's comments suggest, we failed to sufficiently explain the technological innovations behind our integration of the DVD optics into HS-AFM for this application. We have revised the manuscript to correct this flaw.

We wish to point out that combining DVD OPU detectors with high-speed AFM scanning, as we have in this work, is novel. An examination of the state of the field suggests that in a number of respects, our implementation exceeds the current state-of-the-art. Our use of 'contact mode' with OPU detection is novel, as are the

much higher imaging rates achieved, and the much larger measurement areas demonstrated. Because of these features, our DVD OPU implementation uniquely enables the large number of measurements of DNA molecules, which in turn makes possible entirely new statistical analyses of molecule conformation, counting rare species, precision length measurements, etc.

While DVD optics were first implemented in AFMs in 1999 [Quercioli], the field of high-speed AFM is much more recent and we present here results collected at a speed 10,000x faster than our nearest DVD optics competitor [Hwu1]. In a previous publication by the lead specialist in the area of optical pick-up (OPU) atomic force microscopy (AFM), E. T. Hwu [1], it has been shown that OPUs can act as a detection system AC mode AFM to image DNA. In Hwu, the AFM image was formulated with scan rates of 0.2 lines/s at pixel dimensions of 256 x 256. In comparison, the method we presented could perform AFM on a unit area to matching pixel densities 10,000 times in the same time compared to the previously reported technique.

In a separate publication [Hwu 2], by the same lead specialist, the researcher outlines the 'high-performance' and 'streamlined' scan rates that can be achieved in tandem with OPU measurements. However, in that published work tip velocities are achieved of up to 40 $\mu\text{m/s}$, compared to our presented method which achieves scan rates of up to 16,000 $\mu\text{m/s}$. In addition, the stage is specified to have a maximum measurement area of up to 17 μm by 17 μm , whereas with our presented method a 2 cm by 2 cm area is typically achievable. This corresponds to a 700,000 times increase in measurement area.

Finally, the previous approaches are not only limited by the speed of the scan methodology but also AFM mode. Previous reports of OPU based AFM have used tapping or AC mode AFM [Hwu 3]. In AC mode AFM, the imaging rate is limited by the oscillation of the cantilever to pixel rates less than ~ 500 kHz. Therefore, it is not possible to access OPUs' ultra-high bandwidth, capable of measurement rates up to several 100s of MHz, for high-speed AFM. Whereas, with our presented method of high-speed contact mode AFM, with no reliance on the resonant frequency, no such mechanical upper limit has yet been found; in our case, the limit is set by the digitization of the detection system. Therefore, this technology presents an exciting and possible opportunity to increase high-speed AFM to even higher physical measurement rates allowing for both better spatial resolution and frame rates.

While this tremendous speed increase is an essential difference, it is not the only one. Our scan stage is 400x faster [Hwu2] and we operate our instrument in a water-layer mediated imaging mode that is unique to our instrument and critical to making measurements at these rates [Picco]. Finally, our instrument specialises in mapping large (mm^2) areas with nanometre resolution and we report here how by combining large-travel coarse positioners and small high-speed scan stage, the technology enables measurements to be rapidly collected from the entire surface of the sample, rather than the much smaller areas typical of AFM [Hwu2]. These design decisions make our instrument stand out against the body of AFM development to date.

Here we demonstrate the potential for high-speed AFM to enable diagnostically relevant genomics measurements. These measurements are only made possible practically through the use of a high-speed AFM. To build a dataset of DNA measurements with the necessary statistical confidence via conventional AFM, or Hwu's DVD optics based instrument, would take weeks or months of continuous imaging.

[Quercioli] - Quercioli, F., *et al.* "Monitoring of an atomic force microscope cantilever with a compact disk pickup." *Review of Scientific Instruments*, 70(9), 3620–3624, 1999

[Hwu1] - Liao, H.S. *et al.* "Operation of astigmatic-detection atomic force microscopy in liquid environments." *Review of Scientific Instruments*, 84(10), 103709–7, 2013

[Hwu2] - Hwu, E.-T. *et al.* "Low-voltage and high-performance buzzer-scanner", *Nanotechnology*, 24, 455503, 2013

[Hwu3] - Hwu, E.-T. *et al.* "Simultaneous detection of translational and angular displacements of micromachined elements", *Nanotechnology*, 91, pp 221908:1-3, 2007

[Picco] - Picco, L., *et al.* "Breaking the speed limit with atomic force microscopy", *Nanotechnology*, 18, 44030–4, 2007

3) line 69, the authors emphasized the stability of the Cas9-sgRNA-DNA complex in the face of the harsh perturbation generated by the fast moving AFM tip. Because the authors used formaldehyde to fix the complex, I don't think the stability is a very significant discovery.

2.4 We have not explained this issue clearly.

The role of formaldehyde crosslinking is to prevent dissociation of Cas9 from DNA during purification, which precedes heating and imaging. We believe that it is very unlikely that Cas9 is covalently attached to DNA during HS-AFM imaging. Our deposition protocol includes 10 min incubation of deposited Cas9-DNA complexes at 120 C followed by slow cooling, whereas formaldehyde crosslinking is reversed completely at 90C for 1 h or at 95C for 30 min (see [Qiagen] and [Kaufmann]). This data suggests the unique properties of Cas9 to form a stable, slowly dissociating tertiary complex with sgRNA and DNA.

We have revised then methods description accordingly

[Kaufmann] - Kaufmann *et al.* - "In organello formaldehyde crosslinking of proteins to mtDNA: Identification of bifunctional proteins", *PNAS*, 97, pp 7772-7777, 2000

[Qiagen] - "QIAamp-DNA-FFPE-Tissue-Handbook", June 2012

4) lines 87-91, I believe that the authors exaggerate the implication of their results.

2.5 We have eliminated the exaggeration.

5) lines 153-174, sites of translocation can be determined accurately, conveniently, and cost-effectively by RT-PCR. I am not convinced about the author's claim that the approach is a viable option for the job.

2.6 We have revised this section of the manuscript in response to comments from Reviewers 2 and 3. We have emphasized that the clinical utility of nanomapping would be in detecting breakpoints for translocations which are not clustered, and thus not easily or cost effectively detectible with PCR or DNA sequencing. We note that this would include the vast majority of translocations.

The BCL2/IGH translocation experiment was intended by us to be a technical demonstration of our 'nanomapping' in this important and broad class of problems. We chose the BCL2/IGH translocation because it is com-

mon in lymphoma, and likely to be present in easily obtained frozen tissue shavings. In our view, this experiment demonstrates the flexibility of the approach. By mapping the breakpoints from seven patient samples, we have established that the protocol functions with clinical material. In doing so, we use very closely spaced markers (tens of nanometers) that are well distinguished -- a capability well beyond what is possible with any form of optical mapping, including super-resolution optical methods (see Figure 4c, for example).

I do see an important application of the author's approach if it can give quantitative information about copy number of a translocated gene (for example, BCL2/IGH below 20 copies) in the presence of normal genes (for example, million copies of BCL2 and million copies of IGH). Unfortunately, the authors have not demonstrated the capability.

2.7 We agree that there are uses for our 'nanomapping' in low copy number sequence quantitation, though the present study makes no attempt to emphasize this capability. It was our intention in this work to establish the basic technical performance of the method (label-to-label resolution, labeling efficiency and scan throughput parameters) that are necessary inputs when evaluating any potential application.

Overall, certainly the manuscript describes sound technical progress and efficacy of the author's approach, but the implication I can appreciate is quite limited. Therefore, I believe the manuscript does not guarantee broad readership. (end)

2.8 We respectfully point out that within the genomics and medical genetics communities, there has been broad interest in new, viable whole genome assays for clinical genomics, reference assembly, GWAS, gene correction, to name a few such applications. Karyotype analysis and FISH are the two mainstay physical mapping methods used today. Optical mapping is a refinement and improvement of these older methods. Our 'nanomapping' achieves critical improvements in resolution, labeling flexibility, and labeling density vs optical mapping and even newer super resolution optical mapping. Limitations of resolution and labeling density/fidelity have prevented optical mapping from replacing FISH and karyotyping. We believe that this study shows that 'nanomapping' has overcome these hurdles.

Reviewer #3 (Remarks to the Author):

The paper by Mikheikin et al demonstrates a novel method for optical DNA mapping using AFM and Cas9. The method is elegant and the data presented demonstrates its usefulness to some extent. However, I have several concerns that limits the general impact of the presented study.

First of all, I think the authors are neglecting recent literature on fluorescence based optical mapping. As an example, Jeffett et al (ACS Nano) show super resolution of nick-labelled DNA in nanochannels with distances down to 300 bp. The upper limit presented of 300 kbp is also a large underestimation.

3.1 We would respectfully point out that we do discuss super resolution techniques and their potential to improve optical mapping in our manuscript (refs 5-7 in the revision). As suggested, we have cited Jeffett et al, which is indeed a step in improving accuracy of optical mapping. Though aware of the work, we did not cite Jeffett et al. previously as we were unable to reconcile the differences in the conclusions and the results described in their Supplementary Materials, though we cited several other recent studies of a similar nature. We note that Jeffett shows data from less than 10 molecules in total, and the key resolution data seems to come from only two molecules. In contrast, our conclusions are based on measurements of nearly 150,000 molecules and are fully documented in the paper and SI material.

More concerning to me is the fact that the longest DNA reported in the paper is 12 kb while the authors claim that the technique works well on all length scales. Fluorescence based optical mapping in the BioNanoGenomics platform reaches megabasepairs and to compete with such a technique this size range should be demonstrated also using this method. This requires gentle sample handling et c. The length scale where traditional sequencing methods generally fail is above 10 kb and this length scale is completely left out of the current study.

3.2 Given the literature, and the similarity between our DNA surface fixation processes and those used in to optical mapping, DNA combing, Fiber FISH, etc., we have felt that it was not necessary replicate the well-known and widely reported experiments related to handling and elongating long DNA to draw the conclusion that our paper draws. We could add data from longer molecules to our paper; however, we would like to avoid spending scarce resources on experiments that will not affect the conclusion, in our opinion.

To explain this rationale, we would first like to distinguish between optical mapping with (1) immobile molecules (e.g., OpGen) vs (2) mobile molecules in nanochannels (e.g., BioNano). Since both technologies use the same bioinformatics (originally developed by us [Mishra 1-5]), it is fairly straightforward to compare their abilities to map a single molecule restriction/nicking/probe/nanoparticle map to a reference, in a natural manner: OpGen-optical mapping, as well as our 'nanomapping', uses the same protocol for stretching a long double stranded molecule on a surface and querying the immobile molecule; they typically work with 200-500kB long molecules (extending to 1 Mb+ in the hands of expert technicians); they are capable of working with smaller molecules, limited by Raleigh limit (in the case of optical mapping). Our 'nanomapping,' by its ability to work with smaller molecules (e.g., PCR products, CDNA, etc.), gives a distinct advantage because of the resulting wider dynamic range as illustrated with the BRCA1 12.5 Kb analysis.

Note that BioNano-optical mapping uses mobile molecules in solution and cannot achieve more than few hundred Kb, as shown in the following article in Nature:

http://www.nature.com/ng/journal/v49/n4/fig_tab/ng.3802_SF1.html?foxtrotcallback=true.

The entropic Gaussian forces working on the immobile molecules poses an unsurmountable upper limit on molecule size. The mega base-pair maps produced by BioNano are obtained by combining maps from multiple overlapping shorter molecules.

When Cas9 is used as a nanoparticle there is no way of discriminating different gene locations from each other. When studying human DNA it is of great importance to put the genes identified into a general context. The authors must demonstrate this possibility to warrant a publication in a broad journal like Nature Nanotechnology. If not the technique is only useful for very specific parts of genomes that are PCR amplified, which limits general use.

3.3 We note that our technique is not PCR dependent; it can analyze molecules isolated by all known methods (hybrid capture, gel extraction, centrifugation, immune-precipitation, etc.). Any sample analyzable with current optical mapping methods can be analyzed with higher resolution using our 'nanomapping.' We use regular PCR and long range PCR in this study for convenience only. [We have added this comment to the manuscript.]

We would also like to point out that the operational considerations for nanoparticle mapping are the same as for optical restriction mapping, and in some cases, for regular electrophoretic restriction mapping. If Cas9 nanoparticles are directed to prevalent but randomly occurring sequences, such as Alu repeats in human, and their locations are measured with enough precision, it is possible to use the particle pattern to uniquely identify the locus under study.

The relevance of physical mapping and its interpretation in genetics (via map alignment, map assembly, variant calling, etc.) have been studied both empirically and theoretically in the pioneering work of Mishra and colleagues [Mishra 1-5], which date back to the mid 1990's. It has been put to practice repeatedly, by us and other colleagues, for a variety of organisms, including human, as well as by the developers of OpGen's and BioNano's version of optical mapping.

In this context, one must address two problems: one, statistical plausibility (governed by 0-1 laws that depend on thresholds in various error sources) and the other, computational complexity (many of the problems are NP-complete, as shown by Mishra et al. in GVOM, II). Both the problems require developing technologies that control the error processes primarily determined by sizing error, accuracy, and labeling rates: certain error sources/ambiguities are inconsequential (e.g., molecule orientation) whereas certain error sources (e.g., molecular chimerism in Opgen technology or systematic breakages in BioNano technology) are unsurmountable.

However, the analysis of these mathematical properties are impossible to prove (or disprove) via just empirical analysis, and are usually accomplished by sophisticated mathematical (e.g., Erdos-Renyi style probabilistic) analysis as described by Mishra in GVOM I. Our main motivation for developing nanomapping was to develop the capabilities that achieve the theoretical bounds (already published) and we discuss these ideas in the paper. Again, few empirical studies will definitely add to the impression of the plausibility of the technology, but will not add to the conclusions that need to be built on sophisticated reasoning that rely on mathematics, computational complexity and technology. These considerations require such technologies to be developed by a multi-disciplinary team, and also make it of broad interest.

[Mishra 1] - "Genomics via Optical Mapping I: 0-1 Laws for Single Molecules," (with T.S. Anantharaman) BIO-RXIV/2013/000844, November 2013.

[Mishra2] - "Genomics via Optical Mapping II: Ordered Restriction Maps," (with T.S. Anantharaman and D.C. Schwartz), *Journal of Computational Biology*, 4(2):91-118, 1997. (In pdf)

[Mishra3] - "Genomics via Optical Mapping III: Contiging Genomic DNA and Variations," (with T.S. Anantharaman and D.C. Schwartz), *Proceedings 7th Intl. Cnf. on Intelligent Systems for Molecular Biology: ISMB '99*, 7:18-27, AAAI Press, 1999.

[Mishra4] - "Genomics via Optical Mapping IV: Sequence Validation via Optical Map Matching," (with M. Antoniotti et al.), CIMS TR-811, Courant Institute of Mathematical Sciences, NYU, March 2001.

[Mishra5] - "Fast and Cheap Genome wide Haplotype Construction via Optical Mapping," (with T.S. Anantharaman and V. Mysore.), *The Pacific Symposium on Biocomputing*; (Eds. R.B. Altman, A.K. Dunker, L. Hunter & T.E.Klein), PSB 2005:385-96, World Scientific, January, 2005.

REVIEWERS' COMMENTS:

Reviewer #1 (Remarks to the Author):

The authors have considered all the reviewers points carefully and added the necessary new information in the manuscript which is now suitable for publication. I agree strongly with the authors that this scanning probe approach will be competitive against nanopores and optical mapping for the applications they highlight. To increase the throughput by 3 to 4 orders of magnitude is a major break-through that demonstrates that force microscopy is now ready to be utilised as a serious tool in genomic research after 30 years of development. Given the complexity of genome structure, I am of the opinion that all these "nano" approaches (AFM, nanopore and optical) will find important mutually exclusive applications in genomics and sequencing, above and beyond the very powerful traditional parallel methods. Moreover, the AFM approach given here can be applied to a myriad of DNA-protein interactions to explore function and regulation of the genome at the molecular level. My original review did not fully emphasise the step change that this approach brings and the manuscript would have been worthy of publication in Nature Nanotechnology. I heartily recommend publication in Nature Communications.

Reviewer #2 (Remarks to the Author):

The authors prepared CRISPR/Cas9 for labeling specifically and efficiently targeted DNA sites, and used AFM combined with DVD optics to localize the sites with a high precision and time-effectiveness. Examples include singly- and doubly-labeled TERT and HER2 constructs as well as BCL2/IGH translocations. Designing CRISPR/Cas9 of high specificity and efficiency (the labeling rate was 90% for perfect sgRNA match sites and 50% at sites with a single mismatch. And off-target binding was quite low.) is not an easy task at all. I appreciate highly such achievements. Also, I am very impressed by the accuracy of their approach (the ability to distinguish two-to-four base pair length difference in PCR amplicons of approximately 200 bp total length, Fig. 3b).

I am pleased to find that the authors made noticeable progress in integrating DVD optics into HS-AFM for the above application. In particular, the new combination can collect data at a speed of 10,000 x faster than their nearest DVD optics competitor, and can examine an area up to 2 cm x 2 cm, corresponding to a 700,000 times increase.

Overall, the manuscript is well-written, the number of data collected is sufficient to support validity of statistical analysis, and the report presents remarkable progress in high speed morphology mapping for the structural variant detection of DNA. I believe the current progress will get great attention from related communities (AFM, gene analysis, and biomedical diagnosis). I strongly believe that the manuscript merits publication in this journal, Nature Communications.

Here are minor issues to be considered before the publication.

1) In order to apply the current approach effectively, DNA samples on mica surface should avoid from entanglement (or overlapping within the same strand). Whereas the authors described various techniques (use of micro or nano-channels, pressure and electrokinetic flow devices, or laser/magnetic tweezers) in the manuscript (lines 206-207), it is necessary to clarify which technique did the authors employ to minimize such occurrence?

2) It will be nice to get information about copy number of a translocated gene in the presence of outnumbered normal genes from the approach. I encourage the authors to discuss extension of the approach in the future in this regard.

3) It is nice to note that the authors examined BCL2/IGH translocation associated with follicular lymphoma for cell line samples as well as tissue samples of patients (Fig. 4). I am curious how the

authors handled heterogeneity of the patient tissue samples. Please explain. (end)

Reviewer #3 (Remarks to the Author):

This paper describes how DNA mapping can be obtained with a high-speed AFM instrument in combination with CRISP/Cas9 labelling. I have reviewed the updated paper and I still have some important concerns and I do not fully agree with the replies by the authors. The authors claim it is "straight-forward" to image DNA molecules that are hundreds of kb in size and still the largest piece of DNA in the paper is only 12.9 kb. The main USP of mapping methodologies is the large fragments of DNA that can be imaged and the long range sequence information can be obtained. For me this is crucial to warrant publication in a journal like Nat Comm. If it is "straight forward I urge the author to include data of this kind. Furthermore, the "needle in haystack" question its not fully adressed. I agree that most of this can eb investigated in detail but to get unique matches to the human genome, DNA pieces much larger than 12.9 kb need to be imaged, in particular if structural variations are to be identified and characterised. I am also not completely satisfied with the use of PCR to amplify the DNA for the lymphoma patients. This oversimplifies things and a more "realistic" sample preparation would be important to demonstrate.

Unfortunately it is beyond my expertise to evaluate if the improved AFM technique demonstrated is of enough general interest to anyway warrant publication in Nat Comm.

Based on the optical mapping data I however think that the paper is better suited for a journal like Scientific Reports.

REVIEWERS' COMMENTS:

Reviewer #1 (Remarks to the Author):

The authors have considered all the reviewers points carefully and added the necessary new information in the manuscript which is now suitable for publication. I agree strongly with the authors that this scanning probe approach will be competitive against nanopores and optical mapping for the applications they highlight. To increase the throughput by 3 to 4 orders of magnitude is a major breakthrough that demonstrates that force microscopy is now ready to be utilised as a serious tool in genomic research after 30 years of development. Given the complexity of genome structure, I am of the opinion that all these “nano” approaches (AFM, nanopore and optical) will find important mutually exclusive applications in genomics and sequencing, above and beyond the very powerful traditional parallel methods. Moreover, the AFM approach given here can be applied to a myriad of DNA-protein interactions to explore function and regulation of the genome at the molecular level. My original review did not fully emphasise the step change that this approach brings and the manuscript would have been worthy of publication in Nature Nanotechnology. I heartily recommend publication in Nature Communications.

1.1 We thank the reviewer for their comments.

Reviewer #2 (Remarks to the Author):

The authors prepared CRISPR/Cas9 for labeling specifically and efficiently targeted DNA sites, and used AFM combined with DVD optics to localize the sites with a high precision and time-effectiveness. Examples include singly- and doubly-labeled TERT and HER2 constructs as well as BCL2/IGH translocations. Designing CRISPR/Cas9 of high specificity and efficiency (the labeling rate was 90% for perfect sgRNA match sites and 50% at sites with a single mismatch. And off-target binding was quite low.) is not an easy task at all. I appreciate highly such achievements. Also, I am very impressed by the accuracy of their approach (the ability to distinguish two-to-four base pair length difference in PCR amplicons of approximately 200 bp total length, Fig. 3b).

I am pleased to find that the authors made noticeable progress in integrating DVD optics into HS-AFM for the above application. In particular, the new combination can collect data at a speed of 10,000 x faster than their nearest DVD optics competitor, and can examine an area up to 2 cm x 2 cm, corresponding to a 700,000 times increase.

Overall, the manuscript is well-written, the number of data collected is sufficient to support validity of statistical analysis, and the report presents remarkable progress in high speed morphology mapping for the structural variant detection of DNA. I believe the current progress will get great attention from related communities (AFM, gene analysis, and biomedical diagnosis). I strongly believe that the manuscript merits publication in this journal, Nature Communications.

Here are minor issues to be considered before the publication.

1) In order to apply the current approach effectively, DNA samples on mica surface should avoid from entanglement (or overlapping within the same strand). Whereas the authors described various techniques (use of micro or nano-channels, pressure and electrokinetic flow devices, or laser/magnetic tweezers) in the manuscript (lines 206-207), it is necessary to clarify which technique did the authors employ to minimize such occurrence?

2.1 In this study, we simply reduced the concentration of the molecules in solution such that overlaps were rare. The elongation was produced by fluid shear at the mica/fluid boundary as the sample solution was pipetted onto the tilted surface, and flowed under the force of gravity. We have added language to the Methods section to clarify this procedure.

2) It will be nice to get information about copy number of a translocated gene in the presence of outnumbered normal genes from the approach. I encourage the authors to discuss extension of the approach in the future in this regard.

2.2 In the *BCL2-IGH* mapping experiments, we did not examine the histology of the lymph node tissues, so we do not know the fraction of B cells harboring the translocation. This sample would be expected to contain a significant number of cancerous cells, all of which would have the translocation. We used long range PCR to prepare the DNA molecules and only the translocated locus would be amplified. Under these conditions, we would expect to detect even relatively rare instances (e.g., cancer cells present at 1% abundance or greater).

For loci that can be effectively sampled with long PCR, or with an ensemble of long PCR amplicons, this strategy is likely to be effective in detecting rare species present at 1% abundance or greater. Hybrid capture methods combined with throughput-optimized hardware and software is likely to be the most generally effective approach. This is in fact the method used for most NGS diagnostic panels and whole exome sequencing assays (e.g., Agilent SureSelect capture baits, etc.). In that case, both mutant and normal sequences would be present in the sample, and thus the limit of detection becomes a function of sampling depth. Given the throughput achieved in this study, it is reasonable to project that loci of several hundreds of kilobases can be sampled at sufficient depth to detect mutants present at 10% fraction.

We have included this discussion in the revised manuscript.

3) It is nice to note that the authors examined *BCL2-IGH* translocation associated with follicular lymphoma for cell line samples as well as tissue samples of patients (Fig. 4). I am curious how the authors handled heterogeneity of the patient tissue samples. Please explain. (end)

2.3 Please see our comments in 2.2, above.

Reviewer #3 (Remarks to the Author):

This paper describes how DNA mapping can be obtained with a high-speed AFM instrument in combination with CRISP/Cas9 labelling. I have reviewed the updated paper and I still have some important concerns and I do not fully agree with the replies by the authors.

The authors claim it is "straight-forward" to image DNA molecules that are hundreds of kb in size and still the largest piece of DNA in the paper is only 12.9 kb. The main USP of mapping methodologies is the large fragments of DNA that can be imaged and the long range sequence information can be obtained. For me this is crucial to warrant publication in a journal like Nat Comm. If it is "straight forward I urge the author to include data of this kind.

3.1 In response to the reviewer's concerns, we have revised our discussion to make clear that molecules longer than 13 kb will be required if nanomapping is to become practically useful for de novo structural variant detection and sequence alignment applications.

As discussed in our previous reply, given the literature, and the similarity between our DNA surface fixation processes and those used in to optical mapping, DNA combing, Fiber FISH, etc., we have felt that the appropriate path for future work would be to incorporate these procedures into our protocol. This will require time and effort, however, and we feel that the current results are sufficiently compelling to warrant publication without delay.

Furthermore, the "needle in haystack" question its not fully addressed. I agree that most of this can be investigated in detail but to get unique matches to the human genome, DNA pieces much larger than 12.9 kb need to be imaged, in particular if structural variations are to be identified and characterized.

3.2 We agree with the reviewer that the 'needle in the haystack' question is of critical importance for whole genome mapping and de novo structural variant detection. This is a very complex question to address rigorously with experiments, however, and beyond the scope of the current study, in our opinion.

While the work already presented in the paper points to techniques that will be necessary to achieve these goal, we have changed the language in the paper lest it gives the immodest impression that we have already solved all the open problems in large-scale genome analysis with the current prototype technology.

I am also not completely satisfied with the use of PCR to amplify the DNA for the lymphoma patients. This oversimplifies things and a more "realistic" sample preparation would be important to demonstrate.

3.3 Please see our comments in reply #2.2, above, on this topic. We agree with the reviewer that more general approaches than PCR are desirable for sample preparation, and this should be a major focus of research going forward. As discussed in the manuscript, it was not our intention to demonstrate a comprehensive assay for BCL2-IGH detection, but rather to use this example to show the general technical capability of nanomapping.

Unfortunately it is beyond my expertise to evaluate if the improved AFM technique demonstrated is of enough general interest to anyway warrant publication in Nat Comm.

3.4 We believe that both the breakthroughs in HSAFM imaging of DNA molecules and the CRISPR/Cas9 nanoparticle labeling approach will be of great interest to the readers of Nature Communications